# Changes in Expression and Function of Placental and Intestinal P-gp and BCRP Transporters during Pregnancy

**DOI:** 10.3390/ijms241713089

**Published:** 2023-08-23

**Authors:** Péter Szatmári, Eszter Ducza

**Affiliations:** Department of Pharmacodynamics and Biopharmacy, Faculty of Pharmacy, University of Szeged, H-6720 Szeged, Hungary; szapeti40@gmail.com

**Keywords:** placenta, intestine, P-glycoprotein, breast cancer resistance protein (BCRP), fetal exposure

## Abstract

ABC transporters are ubiquitous in the human body and are responsible for the efflux of drugs. They are present in the placenta, intestine, liver and kidney, which are the major organs that can affect the pharmacokinetic and pharmacologic properties of drugs. P-gp and BCRP transporters are the best-characterized transporters in the ABC superfamily, and they have a pivotal role in the barrier tissues due to their efflux mechanism. Moreover, during pregnancy, drug efflux is even more important because of the developing fetus. Recent studies have shown that placental and intestinal ABC transporters have great importance in drug absorption and distribution. Placental and intestinal P-gp and BCRP show gestational-age-dependent expression changes, which determine the drug concentration both in the mother and the fetus during pregnancy. They may have an impact on the efficacy of antibiotic, antiviral, antihistamine, antiemetic and oral antidiabetic therapies. In this review, we would like to provide an overview of the pharmacokinetically relevant expression alterations of placental and intestinal ABC transporters during pregnancy.

## 1. Introduction

### 1.1. ATP-Binding Cassette (ABC) Transporters

Drug transporters are found in numerous tissues, and they can affect drug absorption, distribution and excretion [1]. It is well-known that pregnancy has an effect on the physiological mechanisms of the body, which also include the expression of ATP-binding cassette (ABC) transporters. If the expression of ABC transporters is altered during the gestation period, it may influence the pharmacokinetic properties of medications, which can modify the plasma concentration of drugs and fetal exposure [2,3,4].

Since P-glycoprotein (P-gp) was discovered in the middle of the 1970s [5,6], ABC transporters have been in the focus of scientific research. In the last decades, 49 ABC subtypes have been found in humans, and they are classified into seven subfamilies as ABCA, ABCB, ABCC, ABCD, ABCE, ABCF and ABCG. Several studies have shown that these transporters are present in a variety of tissues, such as the brain, gut, kidney, liver, placenta, testis, retina and cancer cells. In general, ABC transporters are globular proteins built up by 1250–1550 amino acids and usually contain two types of domains. One of them is the transmembrane domain (TMD), which provides substrate specificity by six to ten transmembrane alfa helixes. The other type is the nucleotide-binding domain (NBD), which is located at the intracellular side and is responsible for adenosine–triphosphate (ATP) binding and hydrolyzation to obtain energy for the transmembrane transport of the substrate [7,8,9,10].

ABCA transporters mainly mediate lipid transport [11], and ABCD takes part in the transport of different coenzymes, lipids and vitamin B_12_ [12], while ABCE and ABCF have no transport function [10]. Among the seven subfamilies for vertebrates, three transporter protein classes function as drug/xenobiotic efflux pumps, including ABCB, ABCC and ABCG. These transporters have specific expression patterns in different tissues, which are under hormonal, transcriptional and epigenetic regulation. Due to their drug efflux properties, the ABC superfamily is also called multidrug resistance (MDR) transporters [13,14,15]. Some members of the ABCA subfamily have also been linked with chemoresistance in pediatric acute lymphoblastic leukemia therapy [16].

In 2022, the International Council for Harmonization (ICH) of Technical Requirements for Pharmaceuticals for Human Use released their M12 harmonized draft guideline, with the participation of the US Food and Drug Administration, the European Medicines Agency, and the Japanese Pharmaceuticals and Medical Devices Agency. The guideline defines the transporters which need to be tested during the development process of a therapeutic product. In this current situation, among the ABC transporters, if the drug is probably a substrate of P-gp and/or BCRP transporters, in vitro studies are required, but not in the case of multidrug resistance-associated proteins (MRPs) [17]. Therefore, in the following chapters, P-gp and breast cancer resistance protein (BCRP) transporters will be discussed in detail.

P-gp and BCRP transporters are the most famous members of the ABC superfamily. Countless studies deal with their structures, expressions, functions and interactions. As science has progressed, they have been identified in an increasing number of organs and tissues with different expression levels both in humans and animals [18]. It was later discovered their importance in the barrier tissues. In recent decades, several factors have been identified, such as diseases, infections or pregnancy, to influence the expression of these transporters. Moreover, numerous studies examine the efflux functions in protective tissues like the brain, intestine or placenta and their effects on drug absorption and disposition. Relatively quickly, researchers realized that intestinal P-gp and BCRP can affect drug absorption, while the placental P-gp and BCRP influence maternal–fetal distribution [19].

In our review, we would like to summarize the current knowledge on the gestational-age-dependent expression alterations of placental and intestinal P-gp and BCRP transporters during pregnancy from in vivo experiments. We also discuss how the altered transporter expressions might influence fetal exposure.

#### 1.1.1. ABCB Subfamily

The ABCB subfamily is the best-characterized class among ABC transporters thanks to P-glycoprotein, which was discovered in drug-resistant Chinese hamster ovary cells in 1976 [5,6]. More than 10 ABCB transporters are known, and they have different functions in humans. The ABCB1 transporter, better known as P-glycoprotein, mediates various drug transports in different tissues due to its wide range of substrate specificity. ABCB2 and ABCB3 transporters, which are also called antigen peptide transporters (TAP), participate in the adaptive immune system. ABCB4 transporters transport phosphatidylcholine in hepatic tissues instead of drug transport. ABCB6-10 transporters have an important role in iron transport in mitochondria. ABCB11 transporters are mainly expressed in hepatic tissues and function to transport bile salt [7,20].

The best-characterized ABCB transporter is P-gp, due to which it has relevance in chemotherapy. It is also called multidrug resistance transporter 1 (MDR1). The molecular weight of the protein is approximately 170 kDa, thanks to which P-gp is a full transporter, which means that it has two TMDs (consisting of 6-6 transmembrane alfa helixes) and two NBDs [7,21]. The transmembrane alpha helixes of the TMDs form a “funnel” with the two ATP-binding sites. Nowadays, some theories exist about the mechanism of P-gp, but they all have the same ATP-dependent active efflux function against a concentration gradient, with minimal differences. The basis of the mechanism is the conformation switch from inward-facing to outward-facing orientation. The inward-facing structure is the first phase of the transport cycle, which is responsible for drug binding. The binding sites are located in the internal cavity where the inward-facing conformation does not allow access for entry of substrates from outside of the cell and provides unidirectional transport to the extracellular compartment [22]. When ATP binds to the first ATP binding site, it is hydrolyzed to adenosine diphosphate (ADP) and inorganic phosphate (Pi), then provides energy to the transporter for a conformation change. The “funnel” collects all the lipophilic molecules from the lipophilic membrane and pumps them back into the extracellular compartment. After that, the second ATP binds to the second nucleotide binding site. With the second ATP hydrolyzation, the “funnel” has enough energy to reach the original conformation. Another theory is that the “funnel” opens a pore through the membrane, and hydrophilic drugs which reach the intracellular compartment with different transcellular transports will be pumped into the extracellular compartment. These mechanisms play key roles in efflux function [1,23,24] (Figure 1).

P-gp has a wide range of substrate selectivity because many amino acid groups are found in the substrate-specific cavity of the protein, and it provides various connection possibilities of the substrates [1,24]. Thanks to these multiple substrate binding sites, other transporters or enzymes such as BCRP or CYP3A4 have similar substrate specificity as P-gp, which has an important role in the cooperation between transporters and drug-metabolizing enzymes of drug-drug interactions [7,25]. Pharmacologically relevant substrates include antibiotics, HIV protease inhibitors, antihistamines, analgetics and anticancer drugs [26]. Substrate specificity is detailed in Table 1.

In humans, two homolog genes exist, *MDR1* (also called *ABCB1* gene) and *MDR2* (also known as *ABCB4*), but only the *MDR1* gene encodes drug resistance to P-gp [27]. In rodents, there are three encoding genes of P-gp. These are *Mdr1* (or *Mdr1b/Abcb1b*), *Mdr2* and *Mdr3* (or *Mdr1a/Abcb1a*). However, only two genes encode the P-gp phenotype: *Mdr1* and *Mdr3* [20].

P-gp has different expression levels in several tissues such as the intestine, liver, kidney, brain, testis, placenta and eye. Based on anatomical localization, P-gp can play different roles in these tissues, as it is responsible for limiting drug absorption in the intestine. After the drugs reach systemic circulation, P-gp protects sensitive tissues from harmful agents, while the P-gp of the liver and kidney support the elimination process of these xenobiotics by transporting them into the bile, urine or gastrointestinal tract [4].

#### 1.1.2. ABCG Subfamily

The ABCG subfamily is another class of the ABC superfamily, which has efflux properties. To date, five transporters are known in this group: ABCG1, ABCG2, ABCG4, ABCG5 and ABCG8. Four of the five transporters function as sterols (like cholesterol) and lipid efflux transporters in humans [28]. Similarly to the previous transporters, ABCG2 can also transport lipid and steroid hormones, but it is best known for its multidrug resistance properties.

Doyle et al. identified the ABCG2 transporter around the 2000s [29,30], and at the beginning of their research, ABCG2 had a few names, such as ABC transporter of the placenta (ABCP) or mitoxantrone resistance protein (MXR), but the most common and used name is breast-cancer resistance protein (BCRP). After P-gp, the ABCG2 transporter is the second-best-known ABC transporter worldwide. The molecular weight of BCRP is only 70–75 kDa (monomer formation), much less than that of P-gp because it is a half-sized transporter with one TMD and one NBD [31,32]. In order to obtain a functioning ABC transporter, two TMDs and two NBDs are needed to form the substrate “funnel”. To achieve this active structure, BCRP may function as a homodimer that is stabilized by disulfide bridges [33,34]. The dimer formation is approximately 140–150 kDa [32]. As previously mentioned, due to the multiple substrate binding sites, BCRP also has substrate specificity similar to other transporters or enzymes such as P-gp or CYP3A4. BCRP has a wide spectrum of chemotherapeutic drugs, antibiotics, lipophilic conjugates, natural and endogenous compounds and even toxins [1,24,26]. The detailed substrate specificity of BCRP is shown in Table 1.

The physiological and pharmacological role of BCRP is similar to that of P-gp due to the fact that BCRP is highly expressed in pharmacokinetically relevant tissues such as the intestine, brain, placenta, liver, kidney, breast and tumor cells [31]. In humans and rodents, BCRP is encoded by one gene, which is *ABCG2*. BCRP has a crucial role in the intestine in limiting the absorption of various xenobiotics in mice [35]. Moreover, it is also present in barrier tissues such as the blood–placenta barrier and the liver or kidney. As a result, BCRP has a significant effect on drug distribution and elimination processes [7,31,33].

**Table 1 ijms-24-13089-t001:** General information about P-gp and BCRP.

		P-gp	BCRP	References
Names	classification	ABCB1	ABCG2	[7]
	other	MDR1	MXR, ABCP	[13,31]
Molecular weight	monomer formation	170 kDa	70–75 kDa	[21,32]
	dimer formation	-	140–150 kDa	[32]
Encoding genes	human	*ABCB1*	*ABCG2*	[20,33]
	rodent	*Abcb1a (Mdr1a), Abcb1b (Mdr1b)*	*Abcg2*	[20,33]
Substrates	analgetics	morphine, loperamide	-	[36]
	antibiotics	cefoperazone, ceftriazone, erythromycin	ciprofloxacin, nitrofurantoin	[37,38,39]
	antidiabetics	metformin	glyburide	[37,39,40]
	antihistamines	cetirizine, fexofenadine, desloratadine, bilastine	-	[41,42,43]
	antineoplastics	doxorubicin, paclitaxel	mitoxantron, methotrexate, imatinib, paclitaxel	[37]
	antivirals	abacavir, lopinavir, ritonavir, tenofovir	abacavir, lamivudin, zidovudin, lopinavir	[37,44]
	Ca-channel blockers	diltiazem, nifedipine, verapamil	nicardipine	[26]
	H_2_-antagonists	cimetidine, ranitidine	cimetidine	[39,45,46]
	others	rhodamine 123, quercetin	folic acid, quercetin	[26]

## 2. Expression of ABC Transporters during Pregnancy

It is well-known that pregnancy has massive effects on the pharmacokinetics and pharmacology of drugs; moreover, the developing fetus’s health is also a risk factor during pharmacotherapy, so pregnant women form a special group in drug therapy [47]. Besides the general physiological changes, such as decreased gastric acid secretion, slower intestinal motility, reduced plasma albumin concentration, increased metabolizing enzyme activity and higher glomerular filtration rate, the expression of ABC transporters may also be altered during pregnancy, which may affect drug absorption, distribution and excretion. Due to these changes in pharmacokinetic processes, fetal exposure may be modified [2].

Clinical research on pregnant women is challenging because of ethical or technical reasons and the limited availability of tissues [48]. Therefore, preclinical animal in vivo studies (mostly rodents) are widespread instead of in vivo human studies. This review also contains mainly animal experiment data.

### 2.1. Placental ABC Transporters

The first step in the development stage of the placenta is successful implantation, followed by the proliferation of trophoblast cells, which is faster than the embryo growth. As a result of proliferation, the placenta is attached to the uterus. The syncytiotrophoblast layer is formed by cytotrophoblasts cells, which differentiate from trophoblast cells. In humans, only a single syncytiotrophoblast layer divides the fetal and maternal blood compartments, while in rodents, the maternal and fetal milieu are separated by three layers: trophoblast giant cells, syncytiotrophoblast I (maternal blood side) and syncytiotrophoblast II (fetal blood side) [44,48].

The placenta has two main functions during the gestation period. One of them is to provide nutrient and waste exchange transports between the mother and the fetus, which are mainly mediated by solute carrier (SLC) transporters [49]. The other function is to protect the fetus from various endogenous and exogenous xenobiotics which may be harmful to the embryo. These protection mechanisms are mediated by ABC transporters [13].

Since the placenta is also born with the fetus, despite the ethical limitations, there is abundant human data available on the placenta [50], including knowledge about ABC transporters. Several ABC transporters are found in the apical (maternal side) and basolateral (fetal side) membranes of the syncytiotrophoblast, and they are responsible for bidirectional transport [13].

#### 2.1.1. Expression of P-gp during Pregnancy

P-gp was the first discovered placental ABC transporter which was identified as related to fetal protection. P-gp is present already in the beginning phases of gestation and functions as an efflux pump on the apical membrane in the placenta, which means that it pumps its substrates from the fetal side to the maternal circulation by using ATP [13,51]. Due to this mechanism, some drugs cannot reach the fetus. The P-gp expression level changes with gestational age, and this expression pattern may differ between species. Several independent research studies exist related to this topic, and these results show similarities (Table 2).

Mathias et al. found in humans that the expression of P-gp in the early stage of gestation (60–90 days) was high; it drastically decreased for the middle stage of gestation (90–120 days), then it further decreased until term. P-gp mRNA levels have similar expression patterns as protein levels. In the case of 60–90-day-old placentas, mRNA levels were also high, and they decreased until term [52]. Gil et al. obtained data similar to the previous results. They measured high P-gp expression at 13–14 weeks, which also decreased in the middle of the pregnancy (20–25 weeks) and at term (38–41 weeks) [53]. Sun et al. measured highly expressed P-gp levels at 7–13 weeks, which decreased with the progression of gestation. They also revealed that there were no significant differences between cesarean section and vaginal delivery samples [54]. Petrovic et al. found lower protein expression in term placentas than in preterm ones [55]. Overall, the role of decreased P-gp expression with the progression of gestation is presumable to protect the fetus in the early stage of pregnancy, when fetal tissues are underdeveloped (such as the blood–brain barrier) and sensitive to harmful xenobiotics [13]. Based on these results, protein and mRNA levels of P-gp are decreased to a great extent toward term, the consequence of which is that the expression of P-gp depends on gestational age in humans.

As mentioned before, two genes, *Abcb1a* and *Abcb1b* encode P-gp in rodents. Novotna et al. examined the expression of P-gp in rat placentas during the gestation period. They found significant differences between the two gene transcripts. The Abcb1a mRNA level started to increase from gestation day 11 and reached its peak concentration on day 19, then slightly fell until day 22 of gestation. In the case of Abcb1b, the mRNA level started to increase from day 11 until the last day of gestation. They could detect the protein’s presence on gestation day 13, and it reached its maximum level on day 18, then decreased until day 22. This research shows that Abcb1a and Abcb1b have different regulation processes with gestational age [56].

Kalabis et al. examined P-gp expression in mouse placentas. They also determined Abcb1a and Abcb1b mRNA levels. As a result, they found that the placenta had a maximum level of both mRNAs on day 12.5, and then it decreased until term. Similar protein expression was observed in the mRNA [57]. Huixia et al. measured decreased P-gp levels from day 10 until day 19 of gestation, but the mRNA level showed no significant changes [58]. Coles et al. also confirmed that P-gp decreased from the beginning of pregnancy toward delivery, but mRNA levels did not change [59]. These results have great similarities with human data.

#### 2.1.2. Expression of BCRP during Pregnancy

Both humans and rodents express BCRP in large quantities, and it is localized in the apical membrane of the syncytiotrophoblast (syncytiotrophoblast II layer in rats) and the fetal capillary endothelium [60,61]. BCRP’s function and substrate specificity are similar to P-gp because it also supports the barrier function to separate the fetal and maternal circulation and regulate the drug transfer from mother to fetus by active transport [44]. Over the past years, several studies were published related to the expression of BCRP during the gestation period; however, their results are different, which raises questions (Table 3).

In humans, the expression of BCRP changes less with gestational age. BCRP expression was similar in the early stage of gestation (60–90 days) and at term. It decreased in the middle of gestation (90–120 days), but this change was not significant. Similarly, there were no significant alterations in BCRP mRNA levels during gestation [52]. Yeboah et al. found that BCRP mRNA was expressed in the early stages of pregnancy (6–13 weeks) and had no significant changes during the whole period of pregnancy. The protein level was higher at term than in the early stages of pregnancy [61]. They also revealed that there were no significant differences between cesarean section and vaginal delivery samples. Schwabedissen et al. found that the expression of ABCG2 mRNA and protein levels decreased from the early preterm of pregnancy (28 ± 1 weeks) until late preterm (35 ± 3) and then term (39 ± 2) in the placenta [62]. Petrovic et al. obtained similar results as Schwabedissen et al., with lower protein expression in term placentas than in preterm ones [55].

In addition to heterogeneous in vivo human studies, there are several studies with rodents. The results from rats show similarities. Cygalova et al. examined BCRP expression in rat placentas on gestation days 12, 15, 18 and 21. They detected BCRP on day 12, which reached its maximum level on day 15 and then decreased until term [63], while Yasuda et al. found that BCRP mRNA and protein expression declined on day 20 of pregnancy, from day 14 [64]. Wang et al. examined placental BCRP expression in mice. They obtained a similar result as Cygalova et al. in the case of rats. In mice, placental BCRP mRNA and protein were detected on day 10 of pregnancy, reached their maximum on day 15 of pregnancy, then decreased on day 19 [3]. Another mouse experiment was carried out by Kalabis et al. They determined BCRP mRNA and protein levels on days 9.5, 12.5, 15.5 and 18.5 of pregnancy. mRNA levels decreased from day 9.5 until day 18.5, and although they could not determine the protein level from day 9.5 because of technical limitations, they determined it on days 12.5, 15.5 and 18.5. The protein level was higher on day 12.5 of pregnancy than on days 15.5 and 18.5, but it was not significant [65].

All in all, to identify the significant causes of these disparities, more research is required both in humans and rodents. Perhaps it can be a solution to increase the number of samples or revise the different technical and implementation processes, but even interindividual gene variants may influence BCRP expressions [66].

### 2.2. Intestinal ABC Transporters

In the gut, a multilayer structure called the intestinal barrier regulates the absorption rate of different compounds. The first layer is the lumen, which contains various biochemical compounds such as gastric acid or digestive enzymes. Mucus, produced by goblet cells, forms the second layer, which allows small molecules to pass while holding the bigger particles to interact with epithelial cells [67,68]. The third layer is the epithelium built up by epithelial cells (enterocytes, goblet, and tuft cells) with tight connections. The lamina propria is the last layer, which contains blood vessels, lymph vessels, nerves as well as a lot of immune cells and coordinates the immune response before the exogenous agents reach the blood [69,70].

The small intestine is the main place of drug absorption. Nevertheless, to enter the systemic circulation, drugs need to cross the intestinal barrier, which is armed by efflux transporters against exogenous agents [68]. Several ABC transporters were identified in the intestinal epithelial cells, mainly on the apical (expanded by a brush-border membrane) side of the membrane. They limit intestinal absorption by pumping agents back to the lumen, which makes it harder for compounds to enter the systemic circulation [69,71,72]. Therefore, this mechanism could be considered as a phase 0 metabolism process. In human and rat intestinal systems, the expression level of ABC transporters is diverse in the different segments of the intestine [73,74]. Studies have shown that both in humans and rodents, the expression of several ABC transporters increases or decreases from the proximal to the distal part. Moreover, sex-specific alterations in the gastrointestinal tract are also observed [74,75,76].

Gastric acid production and gastrointestinal motility are well-known to decrease during pregnancy, which reduces the duration of transit time and may affect drug absorption [2]. In addition, pregnancy also has an effect on the expression of intestinal ABC transporters. As we previously mentioned, tissue availability is limited, and there are several ethical issues and problems which limit research in the population of pregnant women. Due to this, human data are lacking, so only animal experiments are available (Table 4).

#### 2.2.1. Expression of P-gp in the Small Intestine during Pregnancy

P-gp is present on the apical side of the intestine and has a similar function as in the placenta. Thanks to its active efflux properties, it limits the entry of various xenobiotics into the systemic circulation by pumping them back using ATP. This preventing mechanism is also called the gatekeeper function [73]. Both in humans and rodents, the expression of P-gp increases from the proximal to the distal part, and in these segments, the quantity of P-gp is higher in males [75,77].

There is little data from mice about the expression of P-gp in the small intestine during pregnancy. Moscovitz et al. determined altered P-gp mRNA and protein levels in mice during pregnancy. They found that the Mdr1a mRNA levels decreased on days 14 and 17 of gestation and increased on day 19 of gestation. The expression of P-gp decreased on gestation day 17 by 70% in pregnant mice compared to the non-pregnant group [78]. Mathias et al. also examined the intestinal P-gp protein expression in pregnant mice, but they found no significant differences between pregnant and non-pregnant mice [79].

#### 2.2.2. Expression of BCRP in the Small Intestine during Pregnancy

BCRP is localized on the apical membrane of enterocytes and is responsible for the energy-dependent efflux transport of various products in the gut. Since BCRP is on the apical side, it participates in the gatekeeper function to reduce drug absorption from the lumen as P-gp. In humans, BCRP mRNA levels show a decreasing trend from the proximal to the distal gut segments, while in rodents, the expression of BCRP mRNA increases continuously along the small intestine, then decreases in the large intestine. No significant sex-dependent expression alterations are identified either in humans or in rodents. Little data is available related to the expression of intestinal BCRP during gestation in mice [3,80].

Jamie E Moscovitz et al. also examined, in addition to P-gp, intestinal BCRP mRNA and protein levels in mice during gestation. The results show that on days 14 and 19 of pregnancy, BCRP mRNA expression increased, but the results were not statistically significant, while on day 17 of pregnancy, BCRP mRNA and protein levels decreased significantly compared to the controls [78]. Wang et al., who examined BCRP expression in the placenta, also examined it in pregnant mouse small intestines. They found that BCRP expression increased toward term, but these changes were not significant differences [3].

**Table 4 ijms-24-13089-t004:** Intestinal P-gp and BCRP expression alterations during gestation in different species. gd: gestation day.

Transporter	In Vivo mRNA and Protein Expression Alterations with Gestational Age in the Intestine			Reference
	Species	mRNA Expression	Protein Expression	
P-gp (ABCB1)	mouse	Abcb1a mRNA decreases on days 14 and 17 of gestation and increases on day 19	protein level decreases on gd 17 by 70%	[78]
	mouse	no significant differences between the pregnant and non-pregnant groups		[79]
BCRP (ABCG2)	mouse	higher mRNA level on gd 14 and gd 19, but it is not statistically significant significantly lower mRNA and protein levels on gd 17		[78]
	mouse	increases toward term, but it is not significant		[3]

### 2.3. Effect of Pathological Conditions

It is well-known that several pathological conditions can also affect the expression of ABC transporters by influencing the regulation pathways, such as hormone and mediator levels or epigenetic patterns [81]. In general, research shows that pathological alterations are rather linked to lower transporter expression in the placenta and the intestine [82,83].

It was established in humans and rodents that in various infections, such as viral (polyinosinic/polycytidylic acid (poly (I:C) treatment), bacterial (lipopolysaccharide treatment) and protozoal (malaria) infections, the expression of placental P-gp and BCRP mRNA and protein levels are reduced. Maternal body weight, preeclampsia, hypoxia and emotional distress also alter both P-gp and BCRP mRNA and protein expressions in the placenta [49,82,84].

Based on nonpregnant human and rodent studies, intestinal P-gp is reduced in diabetic conditions (STZ-treated mice), obesity, HIV infections and inflammatory bowel disease. Intestinal BCRP expression levels are decreased in renal failure [83,85]. Unfortunately, no data are currently available on the expression properties of P-gp and BCRP transporters during pregnancy complications in the intestine.

## 3. Pharmacokinetic and Pharmacological Relevance of Expression Alterations during Pregnancy

Nowadays, in numerous developed countries, a significant proportion of pregnant women use any kind of medicine (including prescribed and over-the-counter drugs) for several reasons. Sometimes they have medical complications which must be treated with medicines. On the other hand, they are willing to reduce various symptoms associated with pregnancy or would like to prevent and support their own and their child’s health [86,87,88]. The fetus may need prenatal treatment, and the list of drugs used in pregnancy is increasing from year to year. Among them, several drugs are P-gp or BCRP substrates, and some of them are available over the counter in some countries [40,89]. Common drugs are detailed in Table 5.

As we discussed in the previous chapter, P-gp and BCRP transporters have gestational-age-dependent expression alterations during pregnancy in the placenta and the intestine. Therefore, even normal physiology may influence the extent of fetal exposure to a specific substrate by affecting absorption and disposition processes.

Since the spread of *Mdr1a–/–*, *Mdr1b–/–* and *Abcg2–/–* knock-out animal models, the pharmacokinetic relevance of these transporters in absorption and distribution has been confirmed, just like the pharmacological significance [91,92].

Studies have shown that if P-gp substrates like digoxin, fexofenadine, ivermectin, nelfinavir, paclitaxel or tacrolimus are administrated orally to Mdr1 knock-out mice, the area under curves (AUC) and plasma concentration ratios are increased compared to wild-type mice. In addition, an inverse correlation was obtained between intestinal Mdr1 mRNA expression and the level of talinolol AUC and maximal plasma concentration (Cmax) in humans. Bcrp1 knock-out mice were treated per os with BCRP substrates such as salazosulfapyridine, ciprofloxacin or dietary carcinogens, which results in higher substrate AUCs and plasma levels. Moreover, P-gp or BCRP substrates’ co-administration with inhibitors leads to decreased efflux function with increased drug absorption in the intestine [93]. Smit et al. used Mdr1a/1b knock-out pregnant mice to evaluate the placental P-gp role in the distribution process. They treated mice intravenously with P-gp substrates such as digoxin, saquinavir and paclitaxel. The results showed that the fetal plasma concentration ratios of digoxin, saquinavir and paclitaxel are increased compared to the wild-type ratios. Those Abcg2 knock-out mice which are treated with glyburide and nitrofurantoin also have higher fetal exposure than the wild-type mice. Furthermore, giving P-gp or BCRP inhibitors to the mother, the placental P-gp or BCRP activity is suppressed [91,94,95]. Lankas et al. exposed *P-gp* (−/−) knock-out pregnant mice to an 8,9 Z photoisomer of avermectin (L-652,280), which is a teratogen. The fetuses of knock-out mice were 100% sensitive to cleft palate, while the wild-type fetuses showed no abnormalities [96]. The limitation of these studies is that the P-gp and BCRP knockout animals are reliable to modeling the substrate penetration across the placenta, but the correlation between the transporter expression and activity is not always significant.

Popova et al. administrated fexofenadine to pregnant rabbits and measured the concentration of it in the fetal liver. The fexofenadine concentration was significantly higher in the fetal liver on gestation days 21 and 28 than on the 14th day. The results are correlated with the reduced P-gp level, which means that decreased P-gp expression during pregnancy leads to lower P-gp activity and increased substrate concentration in the fetal compartment [97].

These data indicate that the intestinal absorption and fetal distribution of substrates depend on the expression or function of P-gp and BCRP transporters. Thanks to these models, considerable evidence is provided that P-gp and BCRP (and also other ABC transporters) contribute to the limitation of drugs in the intestinal barrier, blood–brain barrier, blood–testis barrier and blood–placenta barrier. In general, the substrates of ABC transporters have a binding intensity, so we can distinguish weak and strong substrate–transporter connections. Weak substrates can easily penetrate across the membrane, so using strong P-gp or BCRP substrates in the therapy of pregnant women may be beneficial because fetal exposure may be lower [91,92], except in diseases or coadministration with inhibitors.

When the expression of the intestinal efflux transporter is decreased or blocked by an inhibitor, the absorption of ABC transporter substrates in the intestine may increase, which also increases the drug level [98]. Sometimes the substrate-specific transporter may be saturated by the high substrate concentration, thus the efflux pump function will not be satisfactory, so substrates may reach the protected compartment by transfer through both the surface of the intestine and placenta barrier [81]. In contrast, if transporter expressions are increased or stimulated by an inductor, the intestinal absorption of the substrate may be decreased; thereby, the plasma concentration will be lower [98]. In the case of the placenta, the mechanisms are similar. Reduced ABC transporter expressions or inhibited efflux functions contribute to an inappropriate pump activity, which can lead to higher drug concentrations in the fetus compartment. Increased transporter levels or transporter inducers result in lower fetal exposure [81].

## 4. Conclusions

In summary, significant gestational-age-dependent changes are observed in the expression of intestinal and placental P-gp and BCRP transporters.

Studies have reported increased placental P-gp expressions at the early sensitive stages of gestation in humans and mice, while in the case of rats, placental P-gp is increased until term. BCRP expressions show differences between the different experiments in humans, while it is decreased with gestational age both in rats and mice. Moreover, studies revealed that intestinal P-gp and BCRP expressions depend on gestational age in mice.

Varying expression levels in different gestational ages or diseases can lead to an altered substrate concentration in the fetus compartment, not to mention when efflux transporter inhibitors or inductors are coadministered with the substrates of the same transporter. Due to the altered fetal exposure, unexpected events may appear during treatment. These results suggest that knowledge of P-gp and BRCP expression may be particularly important for P-gp and BRCP substrates in antibiotic, antiviral, antihistamine, antiemetic and oral antidiabetic therapy.

This review sheds new light on pregnancy-related ABC transporters, and this knowledge can be applied to risk estimation and reduction. Moreover, a new perspective is provided on drug formulation development processes and dose settings. These data may contribute to the development of new strategies and guidelines and lead to safer and more efficient pharmacotherapy during pregnancy.

## Figures and Tables

**Figure 1 ijms-24-13089-f001:**
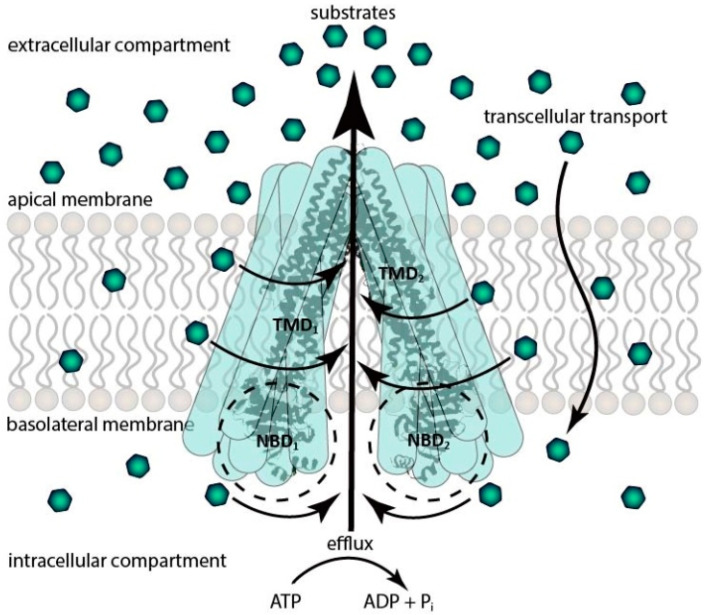
Schematic representation of the efflux pump function of P-gp. TDM: transmembrane domain; NBD: nucleotide-binding domain; ATP: adenosine triphosphate; ADP: adenosine diphosphate.

**Table 2 ijms-24-13089-t002:** Placental P-gp expression alterations during gestation in different species. gd: gestation day.

	In Vivo mRNA and Protein Expression Alterations with Gestational Age in the Placenta			Reference
	Species	mRNA Expression	Protein Expression	
P-gp (ABCB1)	human	high mRNA and protein levels in the early stages, then significant decrease until term		[52]
	human	no mRNA data	high protein level in the early stages, then decrease until term	[53]
	human	mRNA and protein levels decrease with the progression of gestational age no significant differences between cesarean section and vaginal delivery samples		[54]
	human	no mRNA data	significantly lower protein expression in term pregnancies compared to the preterm pregnancies	[55]
	rat	Abcb1a mRNA level increases from gd 11 and reaches its maximum on gd 19 Abcb1b mRNA level increases from gd 11 until gd 22	protein expression reaches its peak on gd 18	[56]
	mouse	Abcb1a/1b levels reach their peak on gd 12.5 then decrease until gd 19	protein level reaches its peak on gd 12.5 then decreases until gd 19	[57]
	mouse	mRNA levels have no significant changes	protein level decreases from gd 10 until gd 19	[58]
	mouse	mRNA levels have no significant changes	protein level decreases from gd 13 until gd 18	[59]

**Table 3 ijms-24-13089-t003:** Placental BCRP expression alterations during gestation in different species. gd: gestation day.

Transporter	In Vivo mRNA and Protein Expression Alterations with Gestational Age in the Placenta			Reference
	Species	mRNA Expression	Protein Expression	
BCRP (ABCG2)	human	mRNA and protein expression levels are similar in the early stages and at term, while in the middle stage of pregnancy, they decrease, but not significantly		[52]
	human	no significant differences between mRNA levels	protein level at 38–41 weeks is higher than at 6–13 weeks no significant differences between cesarean section and vaginal delivery samples	[61]
	human	mRNA and protein levels decrease significantly from early preterm to term		[62]
	human	no mRNA data	significantly lower protein expression in term pregnancies compared to the preterm pregnancies	[55]
	rat	mRNA is detected on gd 12, reaches its peak on gd 15 and then decreases until gd 21	no data	[63]
	rat	mRNA and protein levels decrease from gd 14 to gd 20		[64]
	mouse	mRNA and protein levels are detected on gd 10, reach their peak on gd 15 and then decrease on gd 19		[3]
	mouse	mRNA levels decrease from day 9.5 to day 18.5	protein levels are higher on gd 12.5 than on gd 15.5 and gd 18.5, but not significantly	[65]

**Table 5 ijms-24-13089-t005:** Common drugs are P-gp or BCRP substrates used during pregnancy. * aim to treat the fetus.

Drug	Drug Class	Transporter	Reference
cimetidine	H_2_ antagonist	P-gp and BCRP	[45]
nitrofurantoin	antibiotic	BCRP	[38]
pantoprazole	proton pump inhibitor	P-gp	[40]
fexofenadinedesloratadine	antihistamine	P-gp	[42]
metformin	antidiabetic	P-gp	[40]
metoklopramid	antiemetic	P-gp	[90]
digoxin *	cardiac glycoside	P-gp	[91]
zidovudine *	antiviral	BCRP	[44]

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
