# Peer review of "Changes in Expression and Function of Placental and Intestinal P-gp and BCRP Transporters during Pregnancy"

_ijms, 2023, doi:10.3390/ijms241713089_

Round 1

Author Response

The authors would like to express their thanks to Reviewer #1 for the questions, which have promoted the creation of a better manuscript as concerns its scientific value. Our answers are given below. The changes based on the recommendations of the Reviewers were marked with green color in the revised manuscript.

The expression and function of transporters are pivotal in drug pharmacokinetics. Understanding how these transporters change during pregnancy is vital for ensuring drug efficacy and safety for both the mother and the fetus. This review is specifically centered on the alterations in the expression and function of two crucial drug efflux transporters, placental and intestinal P-glycoprotein (P-gp) and breast cancer resistance protein (BCRP), during pregnancy. However, the authors have not provided a comprehensive understanding of how these changes impact drug pharmacokinetics during pregnancy, leading to a hindrance in the reviewer's enthusiasm. Moreover, the lack of organization in the references and the presence of inaccurately cited sources further affect the credibility of the review.

Thanks for the helpful comments of Reviewer 1, we completed the revised manuscript with more information about the changes in drug pharmacokinetics during pregnancy through these transporters. The numbering of the references was probably due to a software error. These have been corrected in the reised manuscript.

1. Clarify the second sentence on Page 2 (Among…).

We have corrected this in the revised manuscript.

2. PMID: 19325113 is an essential article in the P-gp field.

The manuscript has been supplemented with this article [22].

 3. Ref 24-26 and 30, as well as ref 42-44, were not cited in the main text. Additionally, there is a discrepancy between the last reference in Tab 1 (ref 41) and the first reference in 2. Expression (ref 49), leaving a gap between 42-48. Ref 46 appears after ref 50, and references 42-45 were not cited in the main text.

Thanks for the comment from the Reviewer we corrected these in the text. All references are cited in the main text in ascending order without gaps.

4. Ref 31 is not relevant to BCRP.

The place of this reference (now ref. 37) has been modified.

 5. Most references in 1.1.2 ABCG subfamily are incorrect.

All references in 1.1.2. ABCG chapter has been corrected.

6. On page 4, in the sentence "P-gp or MRPs," "or MRPs" should be deleted as nothing has been mentioned previously.

It was deleted.

7. The statement "BCRP has a crucial role in the intestine in limiting the absorption of various xenobiotics" lacks a supporting reference.

We supported this sentence with reference [36].

8. Tab 5 is split into two pages.

This has changed in the revised manuscript.

9. The section "PK and pharmacological relevance of expression alterations during pregnancy" needs to be expanded and elaborated upon to provide a comprehensive understanding of the topic.

We completed the “Pharmacokinetic and pharmacological relevance of expression alterations during pregnancy” part with new information for better understanding.

Reviewer 2 Report

The manuscript was well written and it is very original and interesting. The abstract should better represent the review and aim of the manuscript. Please reduce the number of keywords.

The introduction is too poor. Authors should improve this part by adding more information on the state of the art of studies on placental and intestinal P-gp and BCRP transporters

The conclusions should be written avoiding redundant information already discussed in the other sections and it should be focused only on the main results in relation to the aims of the study. Morevoer, they should be linked in a better way to the other parts of the paper.

Moderate English revisions are requested.

Author Response

The authors would like to express their thanks to Reviewer #2 for the questions, which have promoted the creation of a better manuscript as concerns its scientific value. Our answers are given below. The changes based on the recommendations of the Reviewers were marked with green color in the revised manuscript.

The manuscript was well written and it is very original and interesting. The abstract should better represent the review and aim of the manuscript. Please reduce the number of keywords.

The abstract was supplemented and the number of keywords was reduced based on the recommendation of the Reviewer.

The introduction is too poor. Authors should improve this part by adding more information on the state of the art of studies on placental and intestinal P-gp and BCRP transporters.

We have expanded the introduction part (1-5 pages) of the revised manuscript with more information about the P-gp and BCRP transporters.

The conclusions should be written avoiding redundant information already discussed in the other sections and it should be focused only on the main results in relation to the aims of the study. Moreover, they should be linked in a better way to the other parts of the paper.

We have modified and supplemented the conclusion part.

Comments on the Quality of English Language: Moderate English revisions are requested.

We made corrections on the spelling grammar and English typing and the manuscript has been reviewed by an English proofreader.

Reviewer 3 Report

In this manuscript, the authors summarized recent results of the changes in expression of placental and intestinal P-gp and BCRP transporters during pregnancy. A general discussion of the ABC transporters is given, followed by a more detailed discussion of the expression and function change of P-gp and BCRP during pregnancy in human and rodents. The manuscript is well organized, and can be a very useful reference for researchers. Several minor issues need to be addressed before the acceptance of this manuscript:

1. On Page 2, Line 14, MRP needs to be defined.

2. Title of the table should be above the table, unless required by the journal.

Author Response

The authors would like to express their thanks to Reviewer #3 for the remarks, which have promoted the creation of a better manuscript as concerns its scientific value. Our answers are given below. The changes based on the recommendations of the Reviewers were marked with green color in the revised manuscript.

In this manuscript, the authors summarized recent results of the changes in expression of placental and intestinal P-gp and BCRP transporters during pregnancy. A general discussion of the ABC transporters is given, followed by a more detailed discussion of the expression and function change of P-gp and BCRP during pregnancy in human and rodents. The manuscript is well organized, and can be a very useful reference for researchers. Several minor issues need to be addressed before the acceptance of this manuscript:

  1. On Page 2, Line 14, MRP needs to be defined.

We defined the MRPs (multidrug resistance-associated proteins) in the revised manuscript.

  1. Title of the table should be above the table, unless required by the journal.

Thank you for your suggestion, we have modified them.